# Interaction Energy Dependency on Pulse Width in ns NIR Laser Scanning of Silicon

**DOI:** 10.3390/mi14010119

**Published:** 2022-12-31

**Authors:** Shunping Li, Xinchang Wang, Guojie Chen, Zhongke Wang

**Affiliations:** 1Guangdong-Hong Kong-Macao Intelligent Micro-Nano Optoelectronic Technology Joint Laboratory, School of Physics and Optoelectronic Engineering, Foshan University, Foshan 528220, China; 2Singapore Institute of Manufacturing Technology (SIMTech), A*Star, 2 Fusionopolis Way, Singapore 138634, Singapore; 3Genuine Solutions Pte Ltd., 80 Ubi Ave 4, #04-02, Singapore 408831, Singapore

**Keywords:** silicon wafer, pulse width, laser ablation, ablation threshold, surface melting

## Abstract

Laser ablation of semiconductor silicon has been extensively studied in the past few decades. In the ultrashort pulse domain, whether in the fs scale or ps scale, the pulse energy fluence threshold in the ablation of silicon is strongly dependent on the pulse width. However, in the ns pulse scale, the energy fluence threshold dependence on the pulse width is not well understood. This study elucidates the interaction energy dependency on pulse width in ns NIR laser ablation of silicon. The level of ablation or melting was determined by the pulse energy deposition rate, which was proportional to laser peak power. Shorter pulse widths with high peak power were likely to induce surface ablation, while longer pulse widths were likely to induce surface melting. The ablation threshold increased from 5.63 to 24.84 J/cm^2^ as the pulse width increased from 26 to 500 ns. The melting threshold increased from 3.33 to 5.76 J/cm^2^ as the pulse width increased from 26 to 200 ns, and then remained constant until 500 ns, the longest width investigated. Distinct from a shorter pulse width, a longer pulse width did not require a higher power level for inducing surface melting, as surface melting can be induced at a lower power with the longer heating time of a longer pulse width. The line width from surface melting was less than the focused spot size; the line appeared either as a continuous line at slow scanning speed or as isolated dots at high scanning speed. In contrast, the line width from ablation significantly exceeded the focused spot size.

## 1. Introduction

The laser ablation of solid materials has been investigated for decades [1,2,3,4,5,6,7,8,9,10,11]. In laser ablation, laser irradiation removes matter from the surface [12,13,14]. The laser ablation of solids with fast processing speed and high precision has been utilized in micromachining [8,15,16]. Various studies have focused on understanding the mechanisms of laser–matter interactions, both experimentally and theoretically [17,18,19,20,21,22,23,24]. The effects of laser pulse width, wavelength, fluence, pulse shape, focus position, and repetition rate on material removal, surface morphology, and ablation rate had been studied for different target materials under different operating conditions [2,6,7,11,17,18,19,20,22,25,26,27,28]. These parameters were found to have profound effects on the interaction processes that start with the absorption of laser energy and end with the removal of the target material for various applications such as surface texturing, drilling, grooving, cutting, etc. [29,30,31].

As a semiconductor material, silicon has been used widely in microelectronics, sensors, solar cell technology, and biotechnological applications [32]. Indeed, the laser ablation of silicon has been studied extensively and continues to be studied for more potential applications. It was found that the shorter the laser wavelength, the lower the ablation threshold [33]; for example, an ns UV laser has a lower ablation threshold compared to ns green and ns NIR lasers [34,35,36,37,38]. For optical absorption, it was revealed some time ago that silicon absorption as a measure of mass ablation efficiency was strongly related to laser intensity, pulse duration, and wavelength [34]. The higher the laser fluence, the stronger the laser–silicon interaction [28]. For example, under green laser irradiation, a silicon surface was just heated without annealing at fluence from 0 up to 0.17 J/cm^2^; annealing took place at fluences from 0.17 to 1 J/cm^2^; the surface was damaged at fluences above 1 J/cm^2^; droplets are produced at 3.5 J/cm^2^; and small clusters of Si atoms and Si ions were detected at 6.0 J/cm^2^ [39]. Laser pulse width is an important parameter. At ultrashort pulses such as fs and ps lasers, the laser ablation threshold is significantly lower than that of an ns pulse irrespective of wavelength. For example, for the ablation of crystalline silicon in air, near-IR 785 nm ps lasers have a threshold of 0.69 J/cm^2^; in contrast, the threshold for deep UV 193 nm ns ArF excimer lasers is 1.9 J/cm^2^ [29]. On surface morphology, when the laser fluence is above the melting threshold, periodical ripple structures on silicon surfaces are easily generated either from overlapped spot scanning [40] or from optical polarization [41] with ps laser pulses, or from rapid solidification of capillary waves and standing acoustic waves under deep ns UV laser irradiation [42]. Surface texturing has been successfully utilized to enhance the efficiency of silicon solar panels by increasing the light absorbance significantly [43]. In addition, increasing the beam spot size and repetition rate can decrease the ablation threshold [44]. Lastly, the ablation atmosphere can play an important role in laser–silicon interaction. For example, ablation in water can significantly reduce the thermal damage and redepositions produced during laser ablation [45,46,47,48]. Ablation with external magnetic field assistance could enhance laser pulse energy coupling into silicon for a higher rate of ablation removal efficiency [22,30,49,50]. Ablation with a thin frost layer assistance could significantly boost ablation efficiency, suppress the recast layer, and reduce the heat-affect zone [51]. Recent comprehensive reviews on the laser ablation of silicon may be found in [32,52,53]. 

Due to the thermal effect induced by laser ablation, laser–material interaction and its associated outcomes naturally depend on the pulse duration. Ablation with ultrashort laser pulses is characterized by minimal thermal effects and the formation of a minimal or negligible heat-affected zone [2,54,55]. Ultrafast (ps or fs) lasers can produce a plasma of hot electrons with sufficient energy to break chemical bonds resulting in ablation [56,57]. When the pulse duration (tens of ns and above) is comparable with the heat diffusion time, the laser energy is transferred to the target material causing local heating before vaporization [7,58]. Thus, ultrashort-pulse ablation of silicon has attracted much attention for many years. Many studies have focused on the study of the differences between the processes induced by ultrashort (ps or fs) pulses and those generated by long (ns) pulses [27,59,60,61]. Even within the fs domain, the laser ablation threshold can be very different for different fs pulse duration. The ablation thresholds were determined to be 0.17 and 0.28 J/cm^2^ for pulses with a width of 25 and 400 fs, respectively [62]. Studies on the ps laser irradiation of silicon in recent years revealed that even a few ps difference in pulse width could cause a big difference in ablation threshold [63]. Compared to fs and ps lasers, an ns laser is promising for its wide application due to its low cost and high efficiency [38,52], although ns pulsed laser ablation has the disadvantage of a pronounced thermal effect [7,11,19,26,29,37,38,64,65]. 

Due to the complexity of the processes, the variety of species involved, and the range of length and time scales covered [31], a good understanding of laser ablation mechanisms is challenging both theoretically and experimentally. A general and interesting issue that has not been well understood is the influence of pulse width on the ablation processes within the ns domain. This study aims to explore the interaction energy dependency on pulse width in ns NIR laser ablation of a silicon surface. The focus is on the influence of pulse width on the surface ablation threshold and melting threshold in the laser scanning of silicon. The interplay of the various key processing parameters is to be revealed for the determination of the ablation results during laser scanning.

## 2. Materials and Methods

Commercially available silicon wafers were employed in this study. They had a crystal orientation of (111), a diameter of 60 mm, and a thickness of 650 μm. The silicon wafers were cut into 20 × 20 mm pieces to facilitate laser scanning and sample characterization. 

A commercial fiber laser (SPI: SP-070P-A-EP-Z-B-Y, UK) was employed for the experiments. The laser has a maximum average power output of 70 W, a wavelength of 1064 nm, and a beam quality of M^2^ ≤ 1.6. The laser operates within the pulse frequency range of 1 to 1000 kHz. A highly desirable and convenient feature of this laser is that the pulse width can be tuned in a wide range from 9 to 500 ns through the selection of the pulse waveform. The laser beam was focused through a galvo scanner F-theta lens with a focal length of 160 mm and a focal spot diameter of about 31.54 μm (computed) at 1/e^2^ of the energy peak intensity. 

An identical average laser power or pulse energy under varied pulse widths was achieved through the setting of the power level percentage. For a comparison of the effects of different pulse widths, a fixed pulse frequency of 100 kHz was selected. This was because, if the frequencies were much higher than 100 kHz during laser scanning, the pulse energy was too low to induce sufficient interaction between the laser and the silicon surface. The scanning speed was varied from 50 to 5000 mm/s. The adoption of this wide range of scanning speeds was an attempt to obtain isolated spot separation in a single scan of the laser beam. The isolated spot irradiation on a silicon surface was employed for the investigation of the laser energy thresholds in laser-induced surface ablation and surface melting. During laser irradiation, the average laser power was measured with a power meter (TP50-HP-19, Stellar, Shenzhen, China). After laser irradiation, the surface morphology and scanning line width of the laser-irradiated samples were characterized under an optical microscope (BX53M, Olympus, Tokyo, Japan) in a bright field of vision. 

## 3. Results and Discussion

### 3.1. Interaction Energy Dependence on Pulse Width

Under different pulse widths but at a fixed pulse frequency, the same pulse energy, namely the same average power, could be obtained by adjusting the power percentage level. To have an average power of 4 W, i.e., 0.04 mJ pulse energy at 100 kHz frequency, the power level was adjusted to be about 73, 34, 27, 21, 15, 11, 9.6, 9.6, 9.6, 9.6, and 9.6% for pulse widths of 10, 26, 36, 60, 100, 145, 175, 200, 295, 420, and 500 ns, respectively. (Pa power percentage of 9.6% achieved a power of 4 W for pulse widths of 175, 200, 295, 420, and 500 ns for this laser system.) Figure 1 shows the surface morphology of samples irradiated at a pulse energy of 0.04 mJ under a scanning speed of 50 mm/s, where 0.04 mJ corresponded to an energy fluence of 5.12 J/cm^2^. The silicon surface was strongly ablated at shorter pulse widths of 10 and 26 ns. With a pulse width of 36 ns, the silicon surface was melted instead of ablated. With pulse widths of 60 ns or longer, surface morphology changes were not detected; thus, images for pulse widths of 100 to 500 ns are not shown here. Figure 2 shows the surface morphology of samples irradiated at a higher pulse energy of 0.08 mJ (energy fluence of 10.24 J/cm^2^). The 10 ns pulse width was not investigated as the required 100% laser power output for a pulse energy of 0.08 mJ could not be reached. Clearly, the laser-induced ablation was reduced at increasing pulse width from 26 to 175 ns, with ablation much stronger at a short pulse width of 26 ns. Above 200 ns, surface melting was likely to occur instead of ablation. The level of surface melting decreased with increasing pulse width from 200 to 500 ns. Additional experiments were conducted under higher pulse energies of 0.21, 0.31, and 0.51 mJ by increasing the output average power to 21, 31, and 51 W respectively. The trend in surface morphology changes with pulse width was consistent with previously observed trends. The observed trends for either ablation or melting were similarly applicable for laser irradiation under various scanning speeds. Even under isolated single laser spot irradiation without pulse overlapping under high scanning speeds, surface ablation or melting still depended on the pulse width, as shown in Figure 3 for scanning at 5000 mm/s under a pulse energy of 0.08 mJ (fluence of 10.24 J/cm^2^). These observations indicate that, under the same pulse energy, silicon surface ablation or melting was dependent on the laser pulse width. Shorter pulse widths tend to induce surface ablation, while longer pulse widths tend to induce surface melting. Pulse overlapping or otherwise did not play a significant role in the likelihood of surface ablation or melting with increasing pulse width.

At the same pulse energy but with different time scales, longer pulse widths would result in longer interaction times between the laser and the silicon surface. It should be expected that surface ablation would be more significant at longer pulse widths compared to shorter pulse widths. However, the results obtained were rather counterintuitive, as the likelihood of surface ablation or melting was not determined by the absolute value of the pulse energy but rather strongly related to the time scale, which is to say, the energy deposition rate. 

With the same laser focus, the energy deposition rate may be considered as the average rate of energy available per pulse to be delivered (or deposited) to the sample, i.e., energy deposition in Joules per second for the total pulse energy deposited over the whole pulse width. Thus, the energy deposition rate in every pulse can be simply expressed as: (1)$$\begin{array}{c} E_{\mathrm{rate}}=E_{p}/\tau_{p} \end{array}$$
where $E_{\mathrm{rate}}$ denotes the energy deposition rate (J/s) in every pulse, $E_{p}$ denotes the pulse energy (J), and $\tau_{p}$ denotes the pulse width (s).

At the same time, it should be noted that if there is nonlinear energy absorption by a silicon sample for a given pulse width and energy level, the energy absorbed by the sample will no longer be proportional to the amount of energy delivered/deposited to the sample.

It should be noted that the ratio between energy (J) and time (s) equals power (W), thus the energy deposition rate in Equation (1) refers to a power value. Pulse energy is the total energy from all the photons over the pulse width in a single laser pulse. So, the energy deposition rate in Equation (1) would be an average power value over the whole pulse width. Within a pulse, the instantaneous power changes following the shape of the pulse (namely, following the instantaneous time in the pulse width). The max pulse peak power would correspond to the maximum instantaneous point in the shape of the pulse. The average power in Equation (1) should be proportional to the max peak power. In the literature, pulse energy divided by the time from the start of one pulse to the next is described as laser average power, with the time period covering the pulse on-duty and off-duty times. The pulse energy over the pulse width is taken as laser peak power [66] or pulse peak power [67], with the time period covering the pulse on-duty time only. Hence, Equation (1) may be written as: (2)$$P_{\mathrm{peak}}=E_{p}/\tau_{p}$$
where $P_{\mathrm{peak}}$ represents the laser peak power (W).

The energy deposition rate described in terms of the laser peak power was plotted and is shown in Figure 4. The laser peak power decreased with increasing pulse width. The results show that surface ablation or melting depends on the laser peak power. 

### 3.2. Laser Fluence Threshold Dependence on Pulse Width

A further study has been carried out to understand the threshold in laser fluence for surface melting or ablation under various pulse widths. The ablation threshold can be significantly influenced by the pulse number, which is to say that, the deposition of multiple pulses may reduce the silicon surface ablation threshold due to the incubation effect for ablation in air [44,63,68] or in liquids [47]. Thus, this study focused on the laser fluence threshold of single pulse irradiation of a silicon surface. As the minimum pulse frequency was 1 kHz in our laser system, directly firing the laser beam with a single pulse was not possible. Instead, single pulse irradiation was achieved through scanning of the laser beam at a sufficiently high speed, i.e., at a speed higher than the highest speed at which the focused spots were connected to each other resulting in a continuous line. The maximum scanning speed limit to form a continuous line was calculated by the product of the focal spot diameter and the pulse frequency of 100 kHz, i.e., 3154 mm/s for a focal spot diameter of 31.54 μm. Therefore, a laser beam scanning speed higher than 3154 mm/s would result in isolated irradiation spots. A scanning speed of 4000 mm/s was employed in this investigation. By tuning the laser power, and, thus, the pulse energy, the critical value of laser pulse energy to induce surface ablation or melting only on the silicon wafer surface could be experimentally investigated. 

As an example, Figure 5 shows that for a pulse width of 145 ns, by adjusting the average power percentage, the laser power deposited onto the sample surface could be controlled and measured. This allows the pulse energy fluence to be calculated precisely. Surface ablation or melting was evaluated through surface morphology observation under an optical microscope. As shown in Figure 5, at the higher power level, strong ablation with serious edge thermal effect along the scanning line was produced at the sample surface at power levels exceeding 40%. At power levels between 23 and 22%, no droplet splashing was observed. Only surface melting was observed at a power level of 22%. Thus, a power level of 23% was considered to be the critical value to induce surface ablation. The laser pulse energy fluence for a power level of 23% was thus taken as the ablation threshold. In between 13 and 12%, the irradiation trace induced at the sample surface disappeared and could not be observed; the corresponding laser pulse energy fluence was considered to be the melting threshold. With this experimental protocol, the energy fluence thresholds could be obtained by laser beam scanning of a silicon surface with various pulse widths. 

The threshold values are plotted in Figure 6. The results show that the surface ablation threshold in pulse energy fluence increases with pulse width. Higher power levels, (or high pulse energy) were required to induce surface ablation at longer pulse widths. This result agreed well with the observations in Section 3.1, in which the pulse energy deposition rate indicated by the laser peak power decreased with increasing pulse width. As a result, higher power levels were required to ablate the sample surface. The energy fluence of the ablation threshold increased from 5.63 to 24.84 J/cm^2^ when the pulse width increased from 26 to 500 ns. However, interestingly, the energy fluence threshold to induce surface melting increased at pulse widths shorter than 200 ns. It then remained constant at longer pulse widths up to 500 ns, the longest pulse width investigated. Unlike the ablation threshold, the increase in melting threshold was not significant with increasing pulse width. The energy fluence of the melting threshold increased from 3.33 to 5.76 J/cm^2^ when the pulse width increased from 26 to 200 ns, and then remained constant until a pulse width of 500 ns. This indicated that there was a limiting pulse width value beyond which the melting threshold would remain constant. These results can be attributed to a longer interaction time between the laser pulse and the sample surface for longer pulse widths. Distinct from a shorter pulse width, a longer pulse width did not require a higher power level to induce surface melting. This is because surface melting can be induced at a lower power with the longer heating time of a longer pulse width. Furthermore, unlike laser pulse ablation, the higher laser peak power was always an advantage for ablation with shorter pulses. For a better comparison, the pulse energy fluence at a fixed peak power of 400 W has been plotted in Figure 6 as well. At a peak power of 400 W, the pulse energy fluence linearly increases with pulse width but with most of the values below the ablation threshold. An additional discussion has been provided in Section 3.3 “Identical peak power at various pulse widths”.

### 3.3. Identical Peak Power at Various Pulse Widths 

According to the results in Section 3.2. it can be concluded that laser ablation induced on a silicon surface is strongly correlated to the pulse width as pulse width determines the laser peak power. Shorter pulse widths with higher peak power are advantageous to induce ablation on a silicon surface. Thus, it would be interesting to explore scanning with the same peak power pulses under various pulse widths, i.e., scanning with an identical laser peak power under different interaction times between the laser beam and the silicon surface. 

As an example, a laser peak power of 400 W was applied to scan the silicon wafer sample surface. A peak power of 400 W was obtained by adjusting the average power percentage for various pulse widths with a constant pulse frequency of 100 kHz. For pulse widths of 26, 60, 100, 145, 200, 295, 420, and 500 ns, the power percentages required were 13, 13.4, 15.4, 17.4, 18.4, 20.6, 26.2, 33.4, and 37.4%, respectively. As laser peak power was obtained by dividing pulse energy by pulse width, for a constant laser peak power, the pulse energy would increase with pulse width. This means that the longer the pulse width, the higher the pulse energy fluence. There is a linear relationship between the pulse energy fluence and pulse width, as shown in Equation (3): (3)$$F=E_{p}/A$$

And from Equation (2),
(4)$$E_{p}=P_{\mathrm{peak}}\times \tau_{p}$$
where F denotes the pulse energy fluence (J/cm^2^), A denotes the focal spot area (cm^2^), and $P_{\mathrm{peak}}$ represents laser peak power (here 400 W). 

The detailed energy fluence values for peak power of 400 W at different pulse widths obtained from Equation (3) are plotted in Figure 6. The pulse energy fluences were mostly below the ablation threshold. The pulse energy fluence was below the ablation threshold at pulse widths below 200 ns or near the ablation threshold at pulse widths above 295 ns. The energy fluence was near or below the melting threshold at pulse widths below 100 ns, and above the melting threshold at pulse widths above 145 ns. These results indicate that surface melting should occur at longer pulse widths and no noticeable melting at shorter pulse widths. As shown in Figure 7, at pulse widths below 100 ns, the energy fluence was below the melting threshold and thus no visible melting on the silicon surface was observed by an optical microscope. For a constant laser peak power, the longer pulse widths contain a higher amount of laser energy, and this can be an advantage for laser surface melting or surface ablation. 

When the scanning speed was reduced to a sufficiently low speed, the laser pulses would overlap along the scanning line. The laser melting or ablation threshold was reduced accordingly with pulse overlapping. Surface melting or ablation could occur even when the laser pulse energy fluence was below the fluence threshold, as shown in Figure 8. Surface melting was enhanced under slow speeds with multiple pulses irradiation. This agreed well with previous studies that multiple pulses deposition onto a silicon surface may reduce the surface ablation threshold due to energy incubation [44,63,68]. The correlation of the energy fluence threshold to the number of overlapped pulses was not discussed in this study, as this topic has been well investigated. 

### 3.4. Scanning Line Width at Various Pulse Widths 

From Figure 1, Figure 2, Figure 3, Figure 5, Figure 7 and Figure 8, it can be observed that the scanning width or individual spot crater size was also correlated to the pulse width. The measured scanning line widths under the various pulse widths are summarized in Figure 9. The line width decreased with an increase in pulse width when scanning with a constant pulse energy fluence irrespective of the scanning speed. Comparing the surface morphologies scanned on the silicon surfaces shown in Figure 1, Figure 2, Figure 3, Figure 5, Figure 7, Figure 8 and Figure 9, it was observed that the surface ablation line was larger than the diameter of the focused spot size. In contrast, surface melting produced line widths smaller than the diameter of the focused spot size for either a continuous line at slow scanning speeds or an isolated dotted line at high scanning speeds. 

A schematic illustration in Figure 10 attempts to describe the correlation between the focal spot, scanning line width, and ablation or melting threshold. The line width or isolated spot crater size was smaller than the focal spot size when the pulse energy fluence was below the melting threshold. The line width or isolated spot crater size was larger than the focal spot size when the laser pulse energy fluence exceeded the ablation threshold. At high pulse energies significantly above the ablation threshold, the resulting effective interaction spot crater size was far beyond the focal spot size—for example, when scanning with a pulse energy of 0.31 mJ (39.69 J/cm^2^) and a pulse energy of 0.51 mJ (65.31 J/cm^2^). Under higher pulse energy fluences and slower scanning speeds, the scanning line width was longer. This could be attributed to the slower speed and high energy fluence resulting in a stronger interaction between the laser and the silicon surface, and, thus, multiple pulse depositions in a single spot area and longer interaction time. As a result, the ablation crater size was much larger than the focal spot size, which led to a wider scanning line width, as shown in Figure 11. The thermal edge effect and debris resulting from ablation along the scanning line became more significant with increasing pulse energy fluence. The width values were measured and are plotted in Figure 12. The line width significantly increased with pulse energy fluence when the energy fluence was above the ablation threshold. However, at the same pulse energy fluence, the changes in line width were not significant with a change in pulse width. This could be due to the fact that the focal spot size was constant at the max energy intensity of 1/e^2^. The interplay of the laser parameters needs to be optimized to realize a higher ablation rate and high-quality ablation with less thermal edge effect and debris. Debris was reduced with increasing pulse width, as seen in Figure 2 and Figure 11. A higher ablation rate was achieved with increasing pulse energy fluence as is intuitively shown in Figure 5 and Figure 11. A slower scanning speed is another approach to achieving a higher ablation rate. Overall, shorter pulse width, lower pulse energy fluence, and higher scanning speed would be the preferred laser parameters for achieving high-quality ablation without sacrificing the ablation rate.

## 4. Conclusions

An investigation of surface ablation and melting energy threshold dependency on pulse width in the ns NIR laser scanning of silicon has been carried out. The surface ablation threshold and melting threshold in the laser scanning of silicon were significantly dependent on the pulse width. The level of ablation or melting strongly depended on the pulse energy deposition rate indicated as laser peak power in this study. A shorter pulse width was likely to induce surface ablation; in contrast, a longer pulse width was likely to induce surface melting. Pulse overlapping would not change the likelihood of surface ablation or melting with increasing pulse width. 

The ablation threshold increased from 5.63 to 24.84 J/cm^2^ when the pulse width was increased from 26 to 500 ns. The melting threshold was increased from 3.33 to 5.76 J/cm^2^ when the pulse width was increased from 26 to 200 ns, and then remained constant until a pulse width of 500 ns (the longest pulse width investigated). A higher power level was not required to induce surface melting at longer pulse widths. Under a constant laser peak power, which is to say, a fixed pulse energy deposition rate, the longer pulse width contains a higher amount of laser energy. Thus, a longer pulse width was advantageous for surface ablation.

Surface ablation produced a line width larger than the focused spot diameter. In contrast, surface melting produced a line width smaller than the focused spot diameter, for either a continuous line at slow scanning speeds or an isolated dotted line at high scanning speeds. The line width significantly increased with pulse energy fluence when the energy fluence was above the ablation threshold. The thermal edge effect and debris resulting from ablation along the scanning line were more significant with increasing pulse energy fluence.

## Figures and Tables

**Figure 1 micromachines-14-00119-f001:**
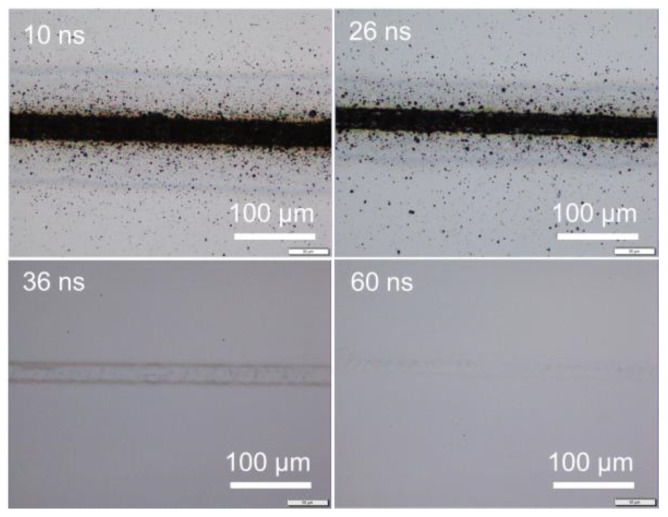
Optical images of irradiated silicon surfaces with 0.04 mJ in pulse energy (5.12 J/cm^2^) at a scanning speed of 50 mm/s.

**Figure 2 micromachines-14-00119-f002:**
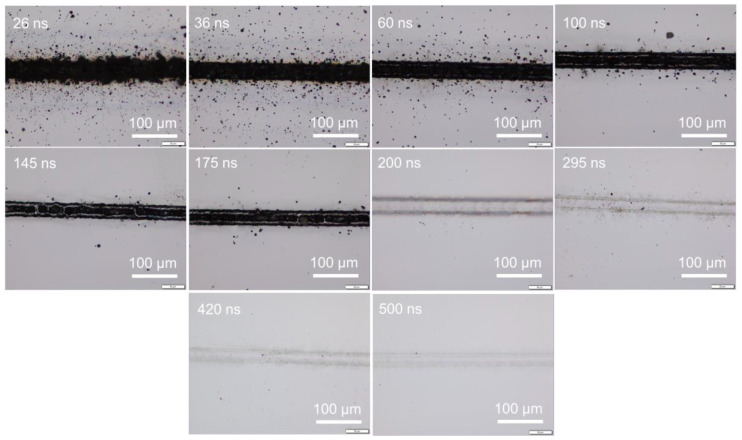
Optical images of irradiated silicon surfaces with 0.08 mJ pulse energy (10.24 J/cm^2^) at a scanning speed of 50 mm/s.

**Figure 3 micromachines-14-00119-f003:**
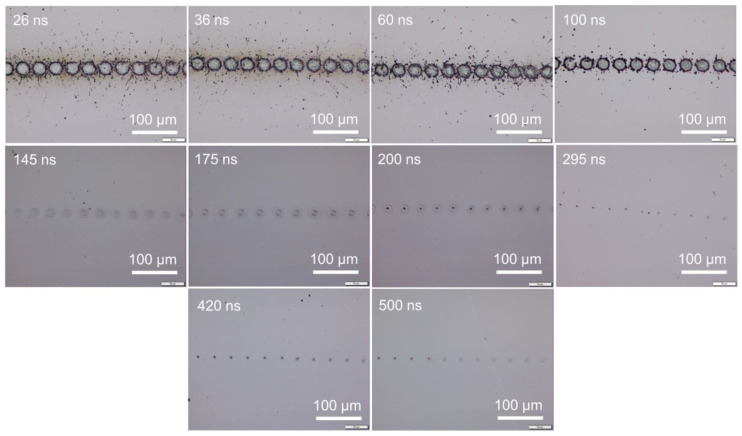
Optical images of irradiated silicon surfaces with 0.08 mJ pulse energy (10.24 J/cm^2^) at a scanning speed of 5000 mm/s.

**Figure 4 micromachines-14-00119-f004:**
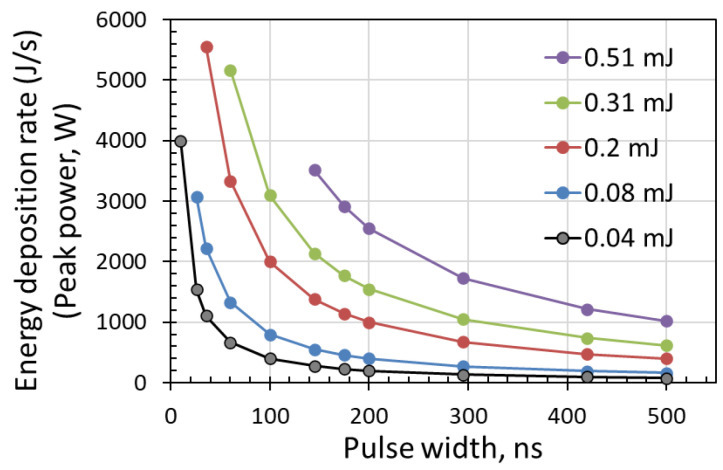
The pulse energy deposition rate in terms of laser peak power versus pulse width at different pulse energies.

**Figure 5 micromachines-14-00119-f005:**
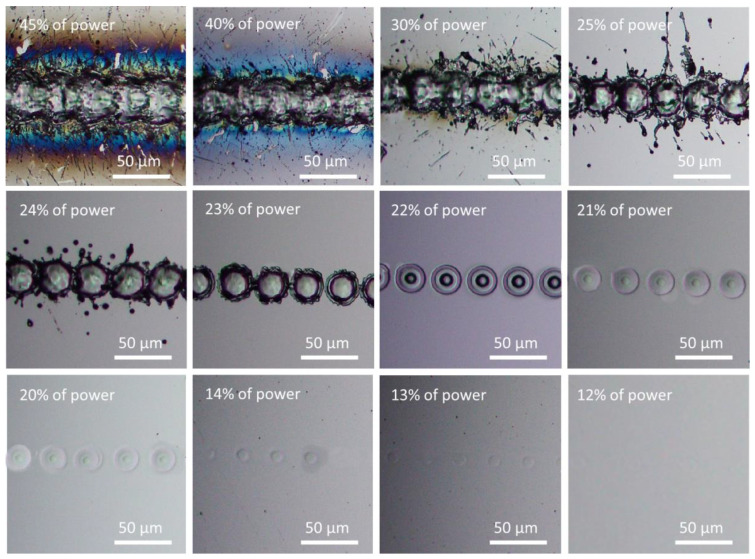
Surface morphologies showing surface ablation or melting under various power levels at 100 kHz and 4000 mm/s scanning speed.

**Figure 6 micromachines-14-00119-f006:**
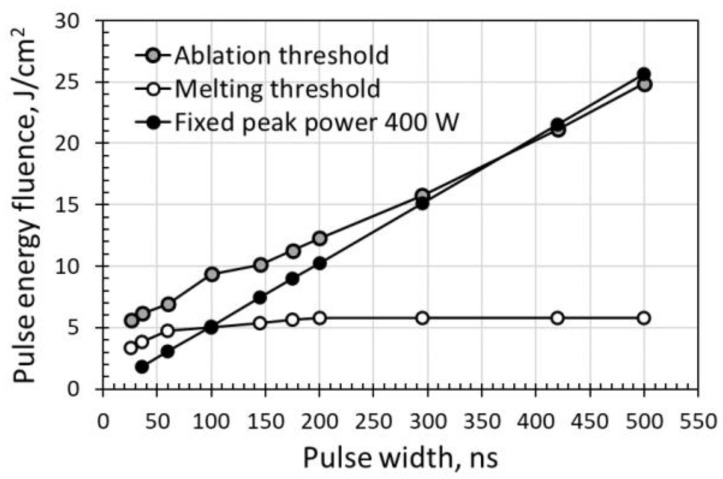
Variation of surface ablation and melting thresholds with pulse width.

**Figure 7 micromachines-14-00119-f007:**
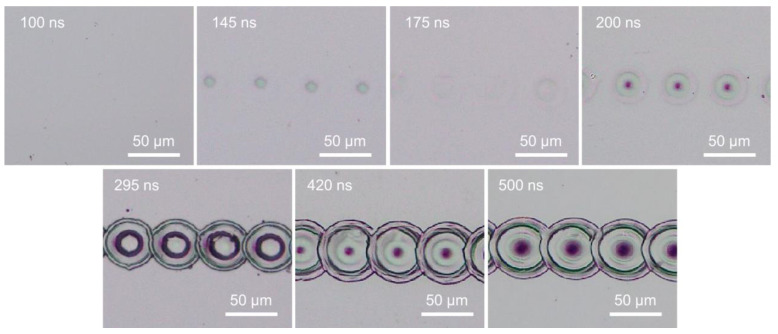
Surface morphologies showing surface melting with increasing pulse width at 4000 mm/s scanning speed.

**Figure 8 micromachines-14-00119-f008:**
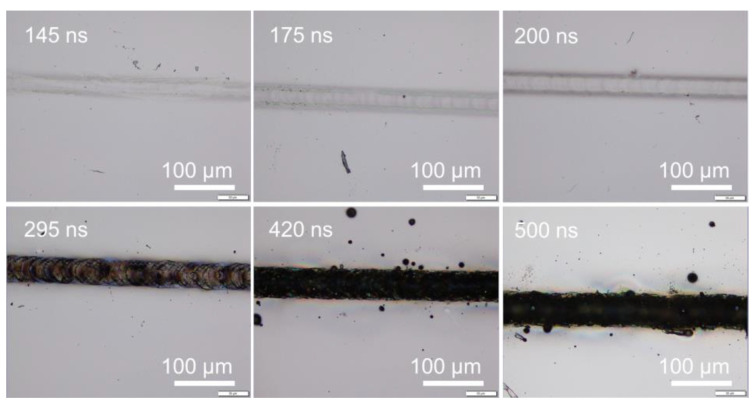
Surface morphologies showing enhanced surface melting with increasing pulse width at 50 mm/s scanning speed.

**Figure 9 micromachines-14-00119-f009:**
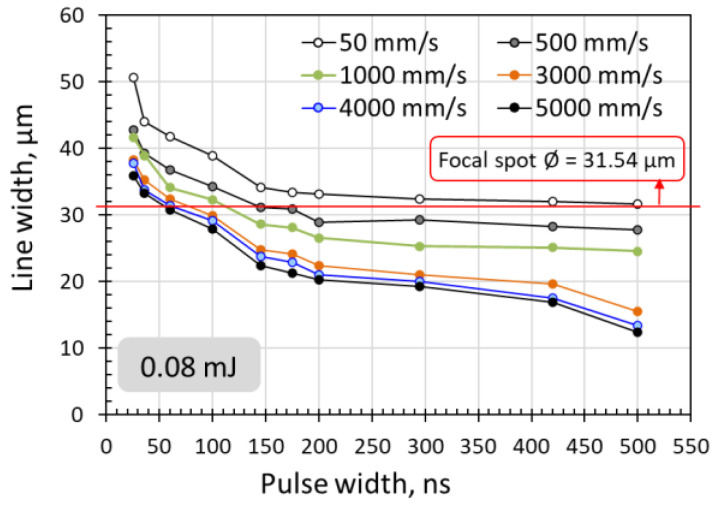
Reduction of scanning line width with increasing pulse width for 0.08 mJ pulse energy (10.24 J/cm^2^) at various scanning speeds.

**Figure 10 micromachines-14-00119-f010:**
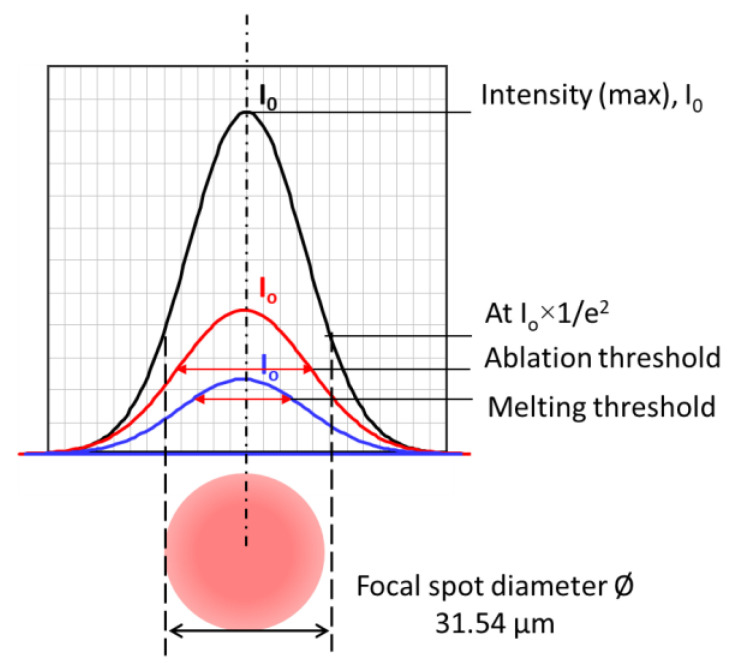
Increase in spot size with pulse energy. Different power percentage *p* gives different maximum energy intensity I_0_.

**Figure 11 micromachines-14-00119-f011:**
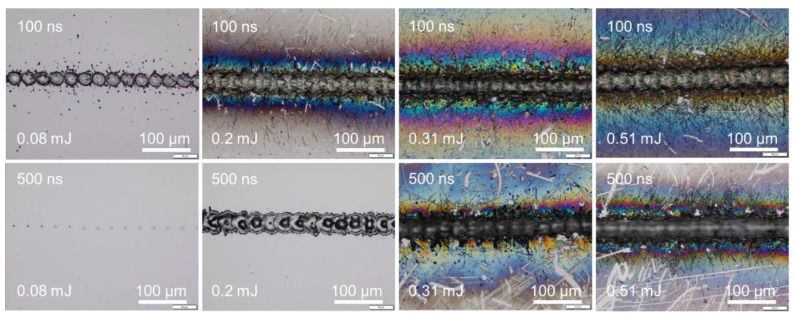
Surface morphologies showing a stronger edge thermal effect and more debris along the scanning line at increasing pulse energy fluence at 400 mm/s scanning speed.

**Figure 12 micromachines-14-00119-f012:**
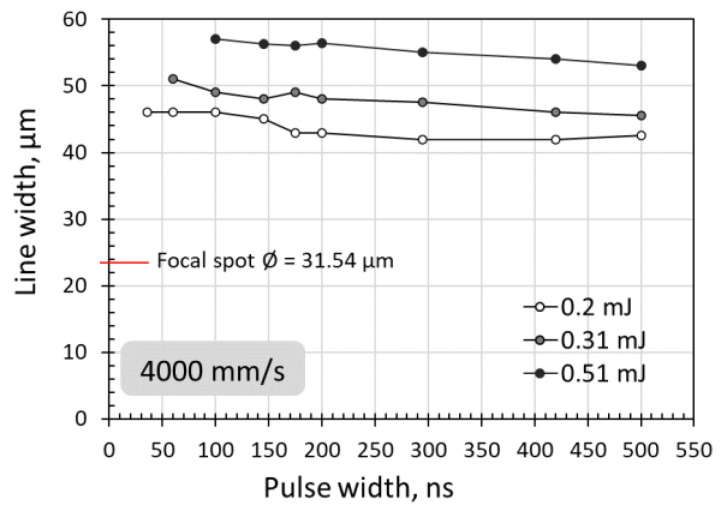
Scanning line width versus pulse energy fluence, showing that there are no significant changes with pulse width at energy fluences far beyond the ablation threshold.

## Data Availability

Not applicable.

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
