# Peer review of "Interaction Energy Dependency on Pulse Width in ns NIR Laser Scanning of Silicon"

_micromachines, 2022, doi:10.3390/mi14010119_

Round 1
Reviewer 1 Report
The authors reported on an investigation of silicon-surface ablation and melting thresholds on the pulse width of nanosecond laser. Although the surface morphologies of surface ablation and melting under various irradiation parameters have been presented in detail, there is still lack of necessary theoretical analysis or explanation on the ablation and melting mechanism. There are several major issues that should be addressed before this manuscript is accepted for publication, as listed below:
(1) The authors proposed a noun called “energy deposition rate”, which is defined by the pulse energy over the pulse width. Actually, the deposition energy on the silicon is not always proportional to the laser pulse energy, and there could be some nonlinear absorption process. Therefore, both the equation 1 and figure 4 are problematical
(2) The theoretical analysis or explanation on the ablation and melting mechanism should be given. For example, a simulation of temperature distribution under laser irradiation with different pulse width could be helpful for understanding the experimental results.
(3) The Figure 7 should be put together with Fig 6 for comparing the pulse energy fluence under a peak power of 400W and the ablation threshold.
(4) Fig. 11 is repeated.
Author Response
Many thanks for the very valuable suggestions to improve our manuscript. Reply to Reviewer 1 (amendments are highlighted blue in the revised manuscript), please see the attached pdf file "Reply to Reviewer 1"

Reviewer 2 Report
The manuscript investigates the ablation and melting threshold with a nanosecond-pulse-scale NIR laser and concludes that longer pulse width can induce surface melting through longer heating time rather than a higher power level. The author also shows how the pulse energy fluence and scanning speed influence the scanning line width. I think the manuscript fits the technical scope of the journal. The authors have given a good elucidation of the impact on the ablation by changing nanosecond NIR laser processing parameters. The experiments support the conclusions, and the work is placed in the proper context. Presentation quality is generally high, as the paper is well-written with clearly-formatted figures. Thus, I recommend acceptance of this manuscript with minor revisions. I have the following questions/suggestions that I hope the authors can address to improve the manuscript: 1. How to realize a higher ablation rate and high-quality ablation with this nanosecond NIR laser? 2. In this paper, the author shows the ablation performance through a lot of figures but does not provide a quantitative description of the ablation volume. I suggest the authors quantify the relationship between laser fluence/pulse width and ablation volume (please consider both the depth and diameter of the ablated area).
Author Response
Many thanks for the reviewer’s encouraging comments and the very valuable suggestions to improve our manuscript. Reply to Reviewer 2 (amendments are highlighted blue in the revised manuscript). Please see the pdf file "Reply to Reviewer 2"

Round 2
Reviewer 1 Report
Thanks much for the authors’s response and additional work.
Some recently work concerned with higher ablation efficiency and better surface quality should be noted, such laser abaltion assisted by water or frost.(see https://doi.org/10.48550/arXiv.2205.09650)
Author Response
Thank you for improving the literature review in the introduction session.
